

# Kondo spectral functions at low-temperatures: A dynamical-exchange-correlation-field perspective

**Zhen Zhao**

Division of Mathematical Physics and ETSF, Lund University,
PO Box 118, 221 00 Lund, Sweden

zhen.zhao@teorfys.lu.se

## Abstract

We calculate the low-temperature spectral function of the symmetric single impurity Anderson model using a recently proposed dynamical exchange-correlation (xc) field formalism. The xc field, coupled to the one-particle Green's function, is obtained through analytic analysis and numerical extrapolation based on finite clusters. In the Kondo regime, the xc field is modeled by an Ansatz that takes into account the different asymptotic behaviors in the small- and large-time regimes. The small-time xc field contributes to the Hubbard side-band, whereas the large-time to the Kondo resonance. We illustrate these features in terms of analytical and numerical calculations for small- and medium-size finite clusters, and in the thermodynamic limit. The results indicate that the xc field formalism provides a good trade-off between accuracy and complexity in solving impurity problems. Consequently, it can significantly reduce the complexity of the many-body problem faced by first-principles approaches to strongly correlated materials.

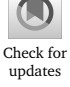

# 1  Introduction

Quantum impurity models (QIMs), where one impurity with a small number of discrete levels is coupled to a noninteracting bath with continuous degrees of freedom, have been extensively studied during the past decades. Originally proposed to study the Kondo effect [1] where a localized spin is screened by conducting electrons due to many-body correlations, QIMs remain to this date the focus of vast interest for their applicability to different topical areas, such as quantum transport through nanoscale devices [2–4], tunneling spectroscopy [5–7], magnetic phase manipulation [8], and many-body entanglement [9, 10]. Moreover, the single-impurity Anderson model (SIAM) [11], one of the basic QIMs variants, is used as an auxiliary system for dynamical mean-field theory (DMFT) [12], a tool in first-principles studies of strongly correlated systems in- and out-of equilibrium [13–15].

Because of this important usage, several types of quantum impurity solvers for the SIAM have been developed. The thermodynamic properties of the SIAM can be exactly solved by the numerical renormalization group (NRG) [16], the continuous-time quantum Monte Carlo (QMC) algorithm [17] and Bethe-Ansatz-based analytic approaches [18–20]. However, the direct application of these solvers to the spectral properties of the SIAM is restricted by factors such as the high computational cost of the original NRG, and the dynamical sign problem or artifacts introduced by the analytic continuation in QMC. Hence advanced solvers arise with sophisticated numerical methods, including generalized NRG [21–24], functional renormalization group [25, 26], configuration interaction approximations [27], distributional exact diagonalization (ED) [28, 29], steady-state density functional theory (DFT) [30–32], expansion QMC [33–35], and non-wave-function-based tensor network approaches [36, 37].

Nonetheless, in spite of these significant advances, there remains a demand for a theoretical treatment of the SIAM which can i) capture spectral weights and energy scales of the Kondo peak and the Hubbard bands in a conceptually and physically transparent way, and ii) be computationally inexpensive in order to make those *ab initio* treatments that use the SIAM as an auxiliary system more numerically affordable.

Recently, a Green's function-based dynamical exchange-correlation (xc) field formalism [38] was proposed. Given the key quantity in the framework, the dynamical xc field (Vxc), the single-particle Green's function, and thus the spectral function, can be solved by a direct integral in the time domain. The Vxc has been calculated exactly for one-dimensional (1D) finite lattice models [39,40] and within the random-phase approximation for the homogeneous electron gas [41]. For those systems, the temporal behavior of the Vxc, $V^{\text{xc}}(t) \sim V_0 + \sum_n A_n e^{-i\omega_n t}$, can be seen as the sum of a constant term (complex for the homogeneous electron gas) plus a small number of oscillating terms accounting for quasiparticle-like excitations. Accordingly, the spectral weight is mainly distributed among a sharp peak (from the constant term $V_0$) and continuous satellite bands that emerge from the oscillating terms in Vxc. Thus, a central task in the approach is to determine the parameters defining Vxc, which naturally implies the introduction of approximate estimates. For example, when applied to 1D half-filled Hubbard lattice and spin-$\frac{1}{2}$ antiferromagnetic Heisenberg lattice at zero temperature, the formalism

approximates the exact lattice Vxc using finite clusters [39, 40]. Consequently, the spectral functions are calculated with a good trade-off between accuracy and computational cost. The quasiparticle-like excitations in the Vxc and the favourable spectral results may be explained by a dynamical screening effect: at zero-temperature, exchange potential and many-body correlations together suppress many degrees of freedom and lead to only few (if not only one) dominant excitations $\omega_n$.

In the local moment regime, some central features of the SIAM local spectral function have an immediate interpretation in terms of the Vxc language. Namely, the Hubbard band and the sharp Kondo resonance peak may be recognised as coming from a 'constant' term and several 'quasiparticle-like' excitation energies. However, the width of the Kondo peak, which is related to the Kondo temperature $T_K$, is exponentially small in the Coulomb interaction $U$, which is different from the Dirac-$\delta$ peak brought by a constant $V_0$. Moreover, in the mixed valence regime, the lack of sharp peak seems to contrast with previous Vxc results from homogeneous lattice models [39, 40]. Therefore, in this paper, we perform a systematic study of the symmetric SIAM from the perspective of the Vxc formalism. Our purpose is two-fold: i) by studying the Vxc of a QIM at finite temperature ($T$), we examine how the low-$T$ thermal excitation and the inhomogeneous setup of the system are reflected in Vxc; ii) we expect to shed new light on the SIAM by investigating the real-time response of the impurity, which is not always easily accessed in conventional self-energy-based approaches. By working directly in real time, it avoids the problem of analytic continuation associated with imaginary-time approaches.

This paper is organized as follows. In Sec. 2, we extend the Vxc formalism, originally proposed for zero-temperature (zero-$T$) systems [38], to finite-temperatures (finite-$T$). This is followed by an application of the developed description to the symmetric SIAM at half-filling, at temperatures upto around the Kondo temperature $T_K$, in Sec. 2.1. In Sec. 3, we first calculate the Vxc analitically on a dimer and numerically on a finite cluster. Based on that, we propose an Ansatz for the SIAM Vxc, from which the local spectral function is obtained. Quantities such as the Kondo temperature and the height of the Kondo peak are re-interpreted within the Vxc framework. Finally, we provide our conclusive remarks and an outlook in Sec. 4.

## 2 Theory

We first derive the general finite-$T$ Vxc formalism and then apply it to a discrete cluster at thermal equilibrium which represents an impurity coupled to a bath. The low-$T$ Vxc is obtained by taking the limit $T \to 0$. We will use atomic units throughout this paper.

For a system with chemical potential $\mu$ at finite temperature $T = 1/\beta$, the generalized time-independent Hamiltonian is

$$\hat{K} = \hat{H} - \mu\hat{N}. \tag{1}$$

With $r = (\mathbf{r}, \sigma)$ the space-spin variable, $\hat{\psi}(r)$ the field operator and $\hat{\rho}(r)$ the density operator, we have

$$\hat{H} = \int dr\, \hat{\psi}^\dagger(r) h_0(r) \hat{\psi}(r) + \frac{1}{2} \int dr\, dr'\, \hat{\psi}^\dagger(r) \hat{\psi}^\dagger(r') v(r, r') \hat{\psi}(r') \hat{\psi}(r), \tag{2}$$

where the single-particle term $h_0(r) = -\frac{1}{2}\nabla^2 + V^{\text{ext}}(r)$ is a sum of kinetic energy and the external field $V^{\text{ext}}$, $v(r, r') = \frac{1}{|\mathbf{r}-\mathbf{r'}|}$ is the Coulomb interaction, and the particle-number operator reads

$$\hat{N} = \int dr\, \hat{\psi}^\dagger(r) \hat{\psi}(r) = \int dr\, \hat{\rho}(r). \tag{3}$$

The Vxc formalism is based on the finite-$T$ time-ordered single-particle Green's function [42]

$$i\bar{G}(rt, r't') := \langle\langle \hat{\psi}(rt); \hat{\psi}^\dagger(r't') \rangle\rangle = \text{Tr}\{\hat{\rho}_G \mathcal{T}[\hat{\psi}(rt)\hat{\psi}^\dagger(r't')]\}, \tag{4}$$

where $\mathcal{T}$ is the real-time time-ordering symbol,

$$\hat{\rho}_G = Z_G^{-1} e^{-\beta \hat{K}}, \tag{5}$$

is the statistical operator,

$$Z_G = \text{Tr}[e^{-\beta \hat{K}}], \tag{6}$$

is the grand canonical partition function, and

$$\hat{\psi}(rt) = e^{i\hat{K}t} \hat{\psi}(r) e^{-i\hat{K}t}, \tag{7}$$

and its conjugate $\hat{\psi}^\dagger$ are the Heisenberg-picture field operators. The $\langle\langle .. \rangle\rangle$ symbol denotes the thermal ensemble average of the time-ordered operators. The equation of motion of the Green's function in the Vxc scheme is

$$[i\partial_t - h(r)]\bar{G}(rt, r't') - V^{\text{xc}}(rt, r't')\bar{G}(rt, r't') = \delta(t-t')\delta(r-r'), \tag{8}$$

where $h(rt) = h_0(r) + V^{\text{H}}(r) - \mu$ contains the Hartree field $V^{\text{H}}(r) = \int dr' v(r, r') \text{Tr}\{\hat{\rho}_G \hat{\rho}(r')\}$. The Vxc is defined according to:

$$V^{\text{xc}}(rt, r't') i\bar{G}(rt, r't') := \int dr'' v(r, r'') \langle\langle \hat{\rho}(r''t)\hat{\psi}(rt); \hat{\psi}^\dagger(r't') \rangle\rangle - V^{\text{H}}(r) i\bar{G}(rt, r't'). \tag{9}$$

We note that for systems in equilibrium, the Vxc in the frequency domain and the self-energy $\Sigma$, defined such that

$$\int dr'' dt'' \Sigma(rt, r''t'')\bar{G}(r''t'', r't') = V^{\text{xc}}(rt, r't')\bar{G}(rt, r't'), \tag{10}$$

are related by the following expression:

$$\frac{1}{2\pi} \int d\omega' V^{\text{xc}}(r, r'; \omega - \omega')\bar{G}(r, r'; \omega') = \int dr'' \Sigma(r, r''; \omega)\bar{G}(r'', r'; \omega). \tag{11}$$

A correlator $g$ can be defined to factorize the high-order term $\langle\langle \hat{\rho}(r''t)\hat{\psi}(rt); \hat{\psi}^\dagger(r't') \rangle\rangle$ [38]:

$$\langle\langle \hat{\rho}(r''t)\hat{\psi}(rt); \hat{\psi}^\dagger(r't') \rangle\rangle = i\bar{G}(rt, r't') g(r, r', r''; t, t')\rho(r''), \tag{12}$$

where $\rho(r'') = \text{Tr}\{\hat{\rho}_G \hat{\rho}(r'')\}$ is the ensemble average of the electron density. We can define a dynamical xc hole

$$\rho^{\text{xc}}(r, r', r''; t, t') = \big[g(r, r', r''; t, t') - 1\big]\rho(r''), \tag{13}$$

which fulfills a sum rule when the number of electrons is conserved (the derivation essentially follows that of the zero-$T$ case [38] except that ground-state expectation value is replaced by thermal average)

$$\int dr'' \rho^{\text{xc}}(r, r', r''; t, t') = -\theta(t'-t), \tag{14}$$

where $\theta$ is the Heaviside step function. Substituting the higher-order term in Eq. (9) with the xc hole, the xc potential can be written as

$$V^{\text{xc}}(rt, r't') = \int dr'' v(r, r'') \rho^{\text{xc}}(r, r', r''; t, t'), \tag{15}$$

which shows that the finite-$T$ xc field can be interpreted as the Coulomb potential of a finite-$T$ xc hole. Furthermore, the xc hole fulfills an exact constraint

$$\rho^{\text{xc}}(r, r', r'' = r; t, t') = -\rho(r), \tag{16}$$

which follows from the fact that the higher-order term $\langle\langle\hat{\rho}(r''t)\hat{\psi}(rt); \hat{\psi}^\dagger(r't')\rangle\rangle$, and thus the correlator $g$, vanishes at $r'' = r$, since $\hat{\rho}(r)\hat{\psi}(r) = \hat{\psi}^\dagger(r)\hat{\psi}(r)\hat{\psi}(r) = 0$. Here we may see an advantage of the Vxc-Framework: the definition of finite-$T$ Vxc introduced here is a natural extension from the zero-$T$ formalism, with ground-state expectation values replaced by thermal ensemble averages. The sum rule and the exact constraint which the xc hole fulfills take the same form as the $T = 0$ case. Moreover, the time-dependence of the external field can be included in a formally straightforward way ($h(r) \to h(rt)$ in Eq. (8), thus $V^{\text{H}}(rt)$ and $\rho(rt)$ depends on time). In practice, however, Vxc can have a more complicated behavior when the system is driven by a time-dependent potential from its ground-state or thermal equilibrium state. The low-$T$ properties of the equilibrium SIAM Vxc is shown in the following section.

## 2.1 Vxc formalism for the SIAM

The SIAM Hamiltonian reads

$$\hat{H}_{SIAM} = \epsilon_f(\hat{n}_{f\uparrow} + \hat{n}_{f\downarrow}) + U\hat{n}_{f\uparrow}\hat{n}_{f\downarrow} + \sum_{k\sigma}\left[\epsilon_k\hat{c}_{k\sigma}^\dagger\hat{c}_{k\sigma} + (v_k\hat{f}_\sigma^\dagger\hat{c}_{k\sigma} + \text{H.c.})\right]. \tag{17}$$

Here $\hat{f}_\sigma^\dagger$ ($\hat{f}_\sigma$) creates (annihilates) an electron with spin $\sigma$ on the impurity site, $\hat{n}_{f\sigma} = \hat{f}_\sigma^\dagger\hat{f}_\sigma$ is the corresponding number operator, $\hat{c}_{k\sigma}^\dagger$ ($\hat{c}_{k\sigma}$) creates (annihilates) a bath electron with energy $\epsilon_k$. Furthermore, $v_k$ is the hybridization amplitude, and $\epsilon_f$ and $U$ are the impurity on-site energy and Coulomb interaction, respectively. We consider a symmetric SIAM at half-filling, which means

$$U + 2\epsilon_f = 0, \tag{18}$$

and the ensemble average

$$n_{f\sigma} = \text{Tr}\{\hat{\rho}_G\hat{n}_{f\sigma}\} = 0.5. \tag{19}$$

We also choose the number of fermionic sites (impurity + bath) $L$ to be even. The local spectral function can be obtained from the impurity Green's function, which can be written in the Lehmann representation as

$$\begin{aligned}
i\bar{G}_{ff,\sigma}(t, \beta) &= \langle\langle\hat{f}_\sigma(t); \hat{f}_\sigma^\dagger(0)\rangle\rangle \\
&= \theta(t)Z^{-1}\sum_{mn_+}e^{-\beta E_m}e^{-i(E_{n_+}-E_m)t}\left|\langle n_+|\hat{f}_\sigma^\dagger|m\rangle\right|^2 \\
&\quad - \theta(-t)Z^{-1}\sum_{mn_-}e^{-\beta E_m}e^{i(E_{n_-}-E_m)t}\left|\langle n_-|\hat{f}_\sigma|m\rangle\right|^2,
\end{aligned} \tag{20}$$

where we set $t' = 0$ since the system is in equilibrium, $m, n_+$ and $n_-$ label eigenstates with $L, L + 1$ and $L - 1$ electrons, respectively, and $Z = \sum_m e^{-\beta E_m}$ is the partition function.

The system has particle-hole symmetry, therefore we focus on the positive time, namely the particle part,

$$i\bar{G}^p_{ff,\sigma}(t>0,\beta) = Z^{-1} \sum_m e^{-\beta E_m} \sum_{n_+} a_{n_+,m;\sigma} e^{-i\omega_{n_+,m}t}, \tag{21}$$

where $\omega_{n_+,m} = E_{n_+} - E_m$ are the excitation energies and $a_{n_+,m;\sigma} = \left|\langle n_+|\hat{f}^\dagger_\sigma|m\rangle\right|^2$ their corresponding weight. The equation of motion of the Green's function reads

$$\left[i\partial_t - \epsilon_f - V^{\mathrm{H}} - V^{\mathrm{xc}}_\sigma(t,\beta)\right]\bar{G}_{ff,\sigma}(t,\beta) = \delta(t), \tag{22}$$

where the Hartree term $V^{\mathrm{H}} = Un_{f\bar{\sigma}}$ is proportional to the density of impurity electron with opposite spin $\bar{\sigma} \neq \sigma$. Here, we emphasize that the Vxc is a result of the Coulomb interaction and can be interpreted as the Coulomb potential of the xc hole, which can be seen from Eq. (15). However, for the SIAM, the hybridization between the impurity and the bath is also a crucial factor influencing the spectral properties. A dynamical hybridization field, also directly coupled to the Green's function in the equation of motion, can be defined within the Vxc-Framework. We incorporate the hybridization field into the Vxc so that the equation of motion has a simpler form, and with a given Vxc, the Green's function can be directly solved. To investigate the hybridization effect, we consider the noninteracting case ($U = 0$ in Eq. (17)). At zero-$T$, the impurity Green's function $G_{ff,\sigma}$ can be analytically solved as

$$G_{ff,\sigma}(\omega) = \frac{1}{\omega - \epsilon_f - \Delta(\omega)}, \tag{23}$$

where

$$\Delta(\omega) = \sum_k \frac{|v_k|^2}{\omega^+ - \epsilon_k}, \tag{24}$$

is the hybridization function. $\Delta(\omega)$ can be calculated analytically by modeling the continuous bath as a tight-binding ring with $N_c$ sites and hopping strength $t_h$, and the impurity site couples to one site with strength $V$ (see Fig. 1). In this model, the SIAM parameters are given by $\epsilon_k = 2t_h \cos(k)$ and $v_k = \frac{V}{\sqrt{N_c}}$. When $|\epsilon_f|, V \ll 2|t_h|$, we approach the so-called wide-band limit (WBL), thus the hybridization function can be treated as a constant for $|\omega| \ll 2|t_h|$,

$$\Delta(\omega) = i\Gamma = i\frac{\pi V^2}{4t_h}. \tag{25}$$

Accordingly, we can solve the hybridization field:

$$V^{\mathrm{xc}}_{\mathrm{nonint,WBL}}(t) = i\Gamma\theta(-t). \tag{26}$$

The physical picture is as follows: the infinitely wide bath band leads to a broadening of the impurity level $\epsilon_f$, which is represented by a purely imaginary hybridization field. This hybridization effect exists also for non-WBL or interacting cases.

## 2.2 The SIAM Vxc at low-temperature

The Vxc coupled to $\bar{G}^p_{ff,\sigma}$ can be obtained from the equation of motion. For the symmetric SIAM at half-filling, $\epsilon_f + V^{\mathrm{H}} = 0$, applying the Lehmann representation of $\bar{G}^p_{ff,\sigma}$ (Eq. (21)) into Eq. (22) and solving for the Vxc, we have

$$V^{\mathrm{xc}}_{p,\sigma}(t,\beta) = \frac{\sum_m e^{-\beta E_m} \sum_{n_+} a_{n_+,m}\omega_{n_+,m}e^{-i\omega_{n_+,m}t}}{\sum_m e^{-\beta E_m} \sum_{n_+} a_{n_+,m}e^{-i\omega_{n_+,m}t}}. \tag{27}$$

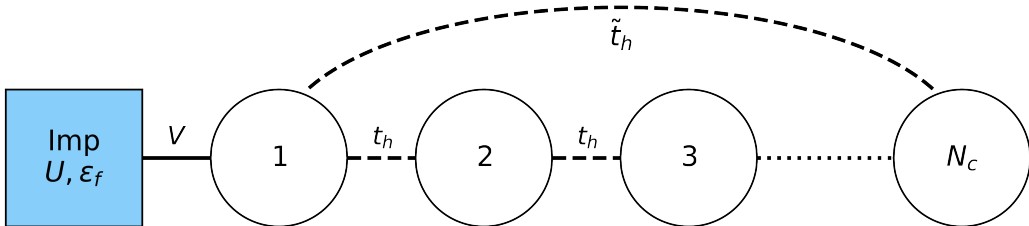

Figure 1: The 1D tight-binding system used to model an impurity coupled to a continuous bath. When periodic boundary conditions are used for the $N_c$ noninteracting sites ($\tilde{t}_h = t_h$), we have the effective SIAM parameters $\epsilon_k = 2t_h \cos(k)$ and $v_k = \frac{V}{\sqrt{N_c}}$. When $\tilde{t}_h = 0$, the model approaches the SIAM setup in the large $N_c$ limit.

We focus on the low-temperature case (referred to as low-$T$) such that $e^{-\beta E_m}$ is negligible except for the lowest two eigenstates $m = 1, 2$. Under this assumption, the Vxc can be written as (the derivation is in Appendix A)

$$V_{p,\sigma}^{\text{xc}}(t, \beta) = V_{p,\sigma}^{\text{xc}}(t, T = 0) + \tilde{V}_{p,\sigma}(t)e^{-\beta(E_2 - E_1)}, \tag{28}$$

which is the zero-temperature $V_{p,\sigma}^{\text{xc}}(t, T = 0)$ plus a correction from a time-oscillating term $\tilde{V}_{p,\sigma}(t)$ and an exponentially small factor. Both the zero-$T$ $V^{\text{xc}}$ and the oscillating term $\tilde{V}$ are determined by the interaction on the impurity site and the hybridization between the impurity and the bath. In the next section, we calculate analytically the low-$T$ Vxc of a dimer where the interaction is nonzero on one site and present the Vxc of an SIAM on a finite cluster determined numerically. Our aim is to investigate the influence of the interaction $U$ and the hybridization $V$ on the Vxc, for the dimer and the cluster, respectively. We will then propose an Ansatz for the finite-$T$ SIAM Vxc and relate the Ansatz parameters to the Kondo physics in the thermodynamic limit.

## 3 Results

### 3.1 Analytic insights from a dimer

We use a dimer with interaction $U$ only on one site and hopping $V$ between the sites to derive the analytic Vxc:

$$\hat{H}_{\text{dimer}} = \epsilon_f(\hat{n}_{f\uparrow} + \hat{n}_{f\downarrow}) + U\hat{n}_{f\uparrow}\hat{n}_{f\downarrow} + V\sum_{\sigma}(\hat{f}_{\sigma}^{\dagger}\hat{c}_{\sigma} + \text{H.c.}), \tag{29}$$

where we fix $\epsilon_f = -\frac{U}{2}$ and the dimer at half-filling. We consider the $T = 0$ case first. We study the large-interaction regime ($U \gg V$) to obtain insights for the SIAM in the Kondo regime. The particle part of Vxc has an approximate form (for the derivation, see Appendix B)

$$V_{p,\sigma}^{\text{xc}}(t, T = 0) \approx \omega_p - \lambda\Omega e^{i\Omega t}, \tag{30}$$

where $\omega_p = \sqrt{\frac{U^2}{16} + 4V^2} + \sqrt{\frac{U^2}{16} + V^2}$, $\lambda \approx \frac{36V^2}{U^2}$, and $\Omega = \sqrt{\frac{U^2}{4} + 4V^2}$. Eq. (30) indicates that the dimer Vxc, similar to the cases mentioned in Sec. 1, can be seen as a sum of a constant term and a quasiparticle-like exponential term. Given the Vxc, the corresponding Green's function can be obtained by solving the equation of motion,

$$\bar{G}_{ff,\sigma}^p(t, T = 0) = \bar{G}_{ff,\sigma}^p(0^+, T = 0)e^{-i(V^{\text{H}} + \omega_p)t}e^{i\int_0^t \lambda\Omega e^{i\Omega\tilde{t}}d\tilde{t}}$$

$$\approx g^+\Big[(1 - \lambda)e^{-i(\epsilon_f + V^{\text{H}} + \omega_p)t} + \lambda e^{-i(\epsilon_f + V^{\text{H}} + \omega_0)t}\Big], \tag{31}$$

where $\omega_0 = \omega_p - \Omega \sim \frac{V^2}{U}$, $g^+ = \bar{G}^p_{ff,\sigma}(0^+, T = 0)$. For the symmetric model at half-filling, $g^+ = -0.5i$ and $\epsilon_f + V^{\mathrm{H}} = 0$. The zero-$T$ spectral function can be obtained with particle-hole symmetry,

$$A_{\mathrm{dimer}}(\omega, T = 0) = \frac{1-\lambda}{2}\delta(\omega + \omega_p) + \frac{\lambda}{2}\delta(\omega + \omega_0) + \frac{\lambda}{2}\delta(\omega - \omega_0) + \frac{1-\lambda}{2}\delta(\omega - \omega_p). \quad (32)$$

Despite the obvious difference in complexity between the dimer and the SIAM, some physics of the SIAM can be outlined from the analytic expression of the dimer Vxc: for large $U$, two peaks ($\omega = \pm\omega_p$) of the spectral function are present, which correspond to impurity levels $\epsilon_f, \epsilon_f + U$. The excitation with energy $\Omega$ creates two central peaks at $\omega = \pm\omega_0 \approx 0$. However, for the dimer the spectral weights of the central peaks, $\frac{\lambda}{2} \sim (\frac{V}{U})^2$, vanish as $U$ increase. The lack of Kondo resonance can be naturally understood as the impurity site is coupled to a single site instead of a continuous bath. This is directly reflected by the Vxc: as $U$ increases, the exponential term (with amplitude $\lambda\Omega \sim \frac{V^2}{U}$) becomes negligible.

For low-$T$, the time-oscillating term in Eq. (28) can be written as

$$\frac{\tilde{V}_{p,\sigma}(t)}{V^{\mathrm{xc}}_{p,\sigma}(t, T = 0)} \approx \lambda' e^{i\Omega' t} - \lambda'' e^{i\Omega'' t}, \quad (33)$$

where $\lambda', \lambda'' \sim \frac{V^2}{U^2}$, $\Omega' \sim U$ and $\Omega'' \sim \frac{V^2}{U}$ (see full expressions in Eqs. (B.27),(B.28), and (B.29) in Appendix B). The Vxc is then

$$V^{\mathrm{xc}}_{p,\sigma}(t, \beta) \approx \omega_p - \lambda\Omega e^{i\Omega t} + e^{-\beta\Delta_0}\omega_p(\lambda' e^{i\Omega' t} - \lambda'' e^{i\Omega'' t}), \quad (34)$$

where $\Delta_0 \sim \frac{V^2}{U}$. Note that we require low temperature condition $e^{-\beta\Delta_0} \ll 1$. The particle spectral function is

$$\begin{aligned}
A_{\mathrm{dimer}}(\omega > 0, \beta) \cong & \frac{1 - \lambda - e^{-\beta\Delta_0}\omega_p(\frac{\lambda''}{\Omega''} - \frac{\lambda'}{\Omega'})}{2}\delta(\omega - \omega_p) + \frac{\lambda}{2}\delta(\omega - \omega_0) \\
& + \frac{e^{-\beta\Delta_0}\omega_p\frac{\lambda''}{\Omega''}}{2}\delta(\omega - \tilde{\omega}_p) - \frac{e^{-\beta\Delta_0}\omega_p\frac{\lambda'}{\Omega'}}{2}\delta(\omega - \tilde{\omega}_0),
\end{aligned} \quad (35)$$

where $\tilde{\omega}_0 = \omega_p - \Omega'$ and $\tilde{\omega}_p = \omega_p - \Omega''$. The first two terms on the RHS of Eq. (35) correspond to the original peaks at zero-$T$, while the last two terms, with weights proportional to $e^{-\beta\Delta_0}$, represent two small peaks (referred to as thermal peaks in the text below) close to the original zero-$T$ peaks, respectively. These thermal peaks arise from expanding the Lehmann representation of the Green's function to the order $e^{-\beta\Delta_0}$. Effectively, the peak at finite-$T$ can be seen as the original peak at zero-$T$ absorbing a thermal peak with close frequencies. Thus the temperature-induced broadening of the SIAM spectral peaks may be explained in the Vxc picture: *at low-$T$, thermal fluctuations induce new peaks close to the original peaks*. The energy difference between the original peak and the thermal peak gives effectively the width of the finite-$T$ spectral peak.

## 3.2 Including hybridization with a finite cluster at zero-$T$

To investigate the combined effects of the interaction $U$ and the hybridization $V$, we numerically solve the Vxc for a finite cluster (corresponding to the $\tilde{t}_h = 0$ case in Fig. 1)

$$\hat{H}_{\mathrm{cluster}} = \epsilon_f(\hat{n}_{f\uparrow} + \hat{n}_{f\downarrow}) + U\hat{n}_{f\uparrow}\hat{n}_{f\downarrow} + V\sum_\sigma(\hat{f}^\dagger_\sigma \hat{c}_{1\sigma} + \mathrm{H.c.}) + \sum_{i,\sigma}^{N_c-1}(t_h\hat{c}^\dagger_{i,\sigma}\hat{c}_{i+1,\sigma} + \mathrm{H.c.}), \quad (36)$$

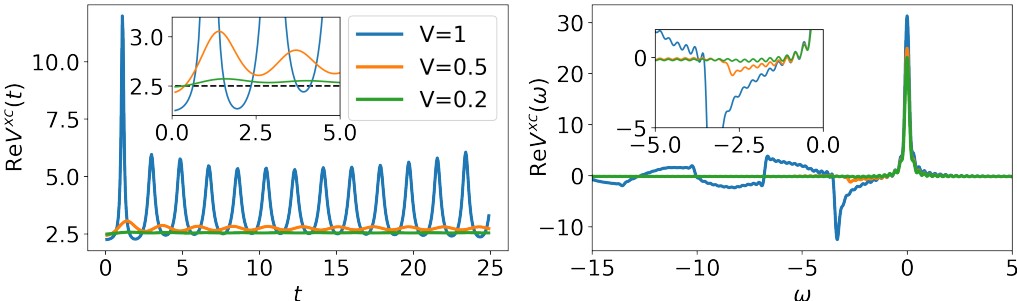

Figure 2: The Vxc of a 50-site cluster at half-filling, with parameters $\epsilon_f = -2.5, U = 5, t_h = -1, T = 0$. Left: real part of $V^{\mathrm{xc}}(t)$. Right: real part of $V^{\mathrm{xc}}(\omega)$, calculated with broadening factor $\eta = 0.1$.

where $N_c$ is the number of noninteracting sites. In the limit $N_c \to \infty$, we reproduce the continuous bath (equivalent to $\epsilon_k = 2t_h \cos(k), v_k = \frac{V}{\sqrt{N_c}}$ in Eq. (17)). We use the ITensor library [43, 44] to perform the time-dependent variational principle (TDVP) [45, 46] algorithm on a symmetric cluster with $L = N_c + 1 = 50$ sites at zero-$T$ (the algorithm performs better in system with open boundary conditions, hence we use a chain setup instead of a periodic one. See detailed treatment in Appendix C). We show in Fig. 2 $\mathrm{Re}V^{\mathrm{xc}}(t)$ with different $V$ values. Also, we plot $\mathrm{Re}V^{\mathrm{xc}}(\omega)$ to analyze the excitation terms contained in the Vxc. For fixed model parameters $\epsilon_f = -2.5, U = 5, t_h = -1$, $\mathrm{Re}V^{\mathrm{xc}}(t)$ with different $V$ can be seen to have three main features. i) It oscillates around some constant values close to $\frac{U+V}{2}$. The constant terms increase with $V$ and correspond to the sharp peaks of $\mathrm{Re}V^{\mathrm{xc}}(\omega)$ at $\omega = 0$. ii) The local maxima of $\mathrm{Re}V^{\mathrm{xc}}(t)$ are approximately equally separated (e.g. for $V = 1$, the time intervals between local maximums are almost 1.85), which may be described by a factor $\mathcal{A}e^{-i\omega_p t}$. The excitation energy $\omega_p$ corresponds to the peaks of $\mathrm{Re}V^{\mathrm{xc}}(\omega)$ at $\omega \sim -3$. Both the amplitude $\mathcal{A}$ and the energy $|\omega_p|$ decrease as $V$ turns smaller, as shown in the inset of Fig. 2 (right panel). iii) For $V = 1$, the local maximum of $\mathrm{Re}V^{\mathrm{xc}}(t)$ around $t = 1$ is much larger than other local maximal values. Correspondingly, $\mathrm{Re}V^{\mathrm{xc}}(\omega)$ exhibits non-Lorentzian structures for $\omega < -5$. Similar larger local maximum at small time also exists for $V = 0.5, 0.2$. As shown in the inset of Fig. 2 (left panel), for $V = 0.5$, the local maximum at $t \sim 2$ is nearly twice as large as the local maximum at $t \sim 4$. This drop in local maxima suggests that $\omega_p$ may contain an imaginary part.

Features i) and ii) can be already found in the analytic expression of the dimer Vxc. However, feature iii) emerges only when the impurity is coupled to a large number of noninteracting sites. Therefore, we attribute feature iii) to the hybridization effect. As mentioned in Sec. 2.1, the Vxc here incorporates the hybridization field, which is a purely imaginary constant for the noninteracting case in the WBL. We note that the constant term of the cluster Vxc has a very small imaginary part. As a result, the spectral function exhibits sharp peaks at $\omega \sim \frac{U}{2}$, instead of proper Hubbard side-bands. This can be attributed to the qualitative difference between the Anderson-type chain with 50 sites and the SIAM where the bath has continuous degrees of freedom.

To summarize the finite-cluster results, $\mathrm{Re}V^{\mathrm{xc}}(t)$ exhibits an oscillating behavior, which suggests that the Vxc can still take the form $V^{\mathrm{xc}}(t) = \mathcal{A}e^{-i\omega_p t} + \mathcal{C}$. However, the hybridization between the impurity and the bath requires a complex $\mathcal{C}$, so that $\mathrm{Re}\mathcal{C}$ and $\mathrm{Im}\mathcal{C}$ determine the peak location and the width of the Hubbard band, respectively. Moreover, the local maxima of $\mathrm{Re}V^{\mathrm{xc}}(t)$ change in time, suggesting a complex $\omega_p$. Here, we notice that $\mathrm{Im}[\omega_p]$ can be positive, leading to $V^{\mathrm{xc}}(t) \to \infty$ for large positive $t$. Thus it is more appropriate to use the form $V^{\mathrm{xc}}(t) = \mathcal{A}e^{-i\omega_p t} + \mathcal{C}$, which is derived using finite clusters, in the small-time regime.

To determine the Vxc in the large-time regime ($|t_l| \rightarrow \infty$), we consider the following observation. The spectral function at low energies is largely determined by $\bar{G}_{ff,\sigma}(t_l, \beta)$. The Kondo resonance peak (with half-width $\Gamma_K$) in the spectral function suggests that the Green's function in the large-time regime takes the form $\bar{G}_{ff,\sigma}(t_l > 0, \beta) \propto e^{-\Gamma_K t_l}$. Since in our formalism $\bar{G}_{ff,\sigma}(t > 0, \beta) \propto e^{-i \int_0^t V^{xc}(\bar{t}) d\bar{t}}$, the Vxc in the large-time regime should be approximately $V^{xc}(t_l > 0) \approx -i\Gamma_K$. Notably, the fact that $V^{xc}$ converges to $-i\Gamma_K$ for large positive $t$ is a direct consequence of the Kondo effect. $V^{xc}$ calculated using small finite clusters lacks this feature because the noninteracting bath is not continuous.

## 3.3 Ansatz of the symmetric SIAM Vxc

Based on the analytic and numerical results above, the particle part of $V^{xc}$ ($t > 0$) of the symmetic SIAM at low-$T$ in the small- and large-time regimes can be described as:

$$
V^{xc}_{p,\sigma}(t > 0, \beta) \approx \left\{ \begin{array}{ll} \lambda \omega_p e^{-i\omega_p t} + \mathcal{C}, & t \text{ small,} \\ -i\Gamma_K, & t \text{ large.} \end{array} \right. \tag{37}
$$

The physical picture is highlighted as follows: $V^{xc}$ in the large-time regime, dominating $\bar{G}_{ff,\sigma}$ for large $|t|$, leads to the sharp Kondo resonance peak. On the other hand, $V^{xc}$ in the small-time regime, with a large contribution from the constant term $\mathcal{C}$, corresponds to the Hubbard side-band broadened by the hybridization effect. We propose an Ansatz for $V^{xc}_{p,\sigma}(t > 0, \beta)$ which captures both the large- and small-time features:

$$
V^{xc}_{p,\sigma}(t > 0, \beta) = \frac{\lambda(\omega_p + \mathcal{C}) + (1-\lambda)\mathcal{C}e^{i\omega_p t}}{\lambda + (1-\lambda)e^{i\omega_p t}}, \tag{38}
$$

where $\lambda$ is real, $\omega_p$ and $\mathcal{C}$ are complex, and $\omega_p + \mathcal{C} = -i\Gamma_K$ is temperature-dependent. The fractional form of the complete Ansatz (Eq. (38)) follows naturally from a Vxc obtained via the equation of motion and the Lehmann representation of the Green's function (see Eq. (27)), where both the numerator and the denominator contain exponential factors of $t$. Note that for a particle-hole symmetric system, the hole part Vxc ($t < 0$) can be calculated using the symmetry relation

$$
V^{xc}(-t) = -V^{xc}(t). \tag{39}
$$

Following Eq. (38), the local Green's function ($t > 0$) is then (see the derivation in Appendix D)

$$
\bar{G}^p_{ff,\sigma}(t, \beta) = -\frac{i}{2} \left[ (1-\lambda)e^{-i\mathcal{C}t} + \lambda e^{-i(\mathcal{C}+\omega_p)t} \right], \tag{40}
$$

and the spectral function is

$$
A(\omega > 0, \beta) = \frac{1-\lambda}{2\pi} \frac{\left| \text{Im}[\mathcal{C}] \right|}{(\omega - \text{Re}[\mathcal{C}])^2 + (\text{Im}[\mathcal{C}])^2} + \frac{\lambda}{2\pi} \frac{\left| \text{Im}[\mathcal{C} + \omega_p] \right|}{(\omega - \text{Re}[\mathcal{C} + \omega_p])^2 + (\text{Im}[\mathcal{C} + \omega_p])^2}, \tag{41}
$$

and $A$ for $\omega < 0$ can be calculated using the particle-hole symmetry:

$$
A(\omega < 0, \beta) = A(-\omega, \beta). \tag{42}
$$

Before determining the Ansatz parameters numerically, we use the Ansatz to interpret the Kondo spectral function. The two peaks in the spectral function can be recognized as a Hubbard side-band located at $\omega = \text{Re}[\mathcal{C}]$ with half-width $\Gamma_H = -\text{Im}[\mathcal{C}]$, and a Kondo peak located at $\omega = \text{Re}[\mathcal{C} + \omega_p] = 0$ with half-width $\Gamma_K = -\text{Im}[\mathcal{C} + \omega_p]$. The spectral weights of the two peaks are determined by $\lambda$. The two peaks have distinct origins. The peak location and the

width of the Hubbard side-band are controlled by the constant term of the Vxc in the small-time regime, which accounts for the fact that the impurity level is affected by the interaction and broadened by the continuous bath. On the other hand, at low-$T$, the Vxc in the large-time regime creates a sharp resonance peak close to $\omega = 0$, whose width and height can be described by the Fermi-liquid treatment [47].

Having in mind the physical meaning of the parameters, we discuss the extrapolation procedure, i.e., how the Ansatz quantities $(\lambda, \omega_p, \mathcal{C})$ can be calculated with a given symmetric SIAM with model parameters $(U, V, t_h, \beta)$. Here, to compare with NRG results in the literature (e.g., from Refs. [29] and [48]), we also use the WBL. We consider first the $T = 0$ limit, and assume that the half-width of the Kondo peak is given by the Kondo temperature $(T_K)$ [48]. Thus,

$$\mathcal{C} + \omega_p \Big|_{T=0} \approx -i T_K = -i \sqrt{\frac{U\Gamma}{2}} e^{-\frac{\pi U}{8\Gamma} + \frac{\pi \Gamma}{2U}} . \qquad (43)$$

The peak location of the Hubbard side-band can be directly calculated, which means

$$\text{Re}[\mathcal{C}] \approx \frac{U}{2} . \qquad (44)$$

We notice that the height of the Kondo peak at $T = 0$ is given by $1/(\pi\Gamma)$, suggesting

$$A(\omega = 0) = \frac{\lambda}{\pi T_K} + \frac{1-\lambda}{\pi |\text{Im}[\mathcal{C}]|} = \frac{1}{\pi\Gamma} , \qquad (45)$$

which simplifies to $\frac{\lambda}{\pi T_K} = \frac{1}{\pi\Gamma}$ for $T_K$ much smaller than the Hubbard side-band half-width $\Gamma_H$, and thus $\lambda$ can be determined. The last unknown parameter is the imaginary part of $\mathcal{C}$, which corresponds to $\Gamma_H$. We use the Anderson-type finite-size chain spectral function to estimate $\text{Im}[\mathcal{C}]$. Note that a finite chain cannot reproduce the proper broadening caused by an infinitely wide band. However, the relative weight between the Hubbard peak and the Kondo peak,

$$R = \frac{1-\lambda}{2\lambda} \frac{T_K}{\left|\text{Im}[\mathcal{C}]\right|} , \qquad (46)$$

can provide information of $\text{Im}[\mathcal{C}]$. We extrapolate the value of $R$ by increasing the number of sites in the chain. We calculate the spectral function using the chain setup with an increasing number of noninteracting sites $N_c$. From the spectral functions (as plotted in Fig. 3a), we determine the relative weight $R$, which is then plotted against the total number of sites $L = N_c + 1$. For a given $\Gamma$, $R$ increases with $L$ (we use $L = 4, 8, 20, 30$ and $40$), as shown by the scattered data points in Fig. 3b. Noticing the nearly linear increase of $R$ at small $L$ and expecting a converging $R$ at large $L$, we fit the $R - L$ data using a hyperbolic-tangent functional form (which eventually provides satisfactory spectral functions) for the fitting function. Consequently, $R(L = \infty)$ can be estimated using the fitting results (see the curves in Fig. 3b).

In Fig. 4, we show the local spectral function of a symmetric SIAM in the WBL with $U = 3, t_h = 50, T = 0$. We choose the parameters ($\Gamma = 0.2, 0.5,$ and $0.9$) to compare with NRG results [29] in the WBL. The spectral function show satisfactory agreements to the NRG results. We attribute this to the fact that the Vxc as an effective field captures the intrinsic physics of an impurity problem. In the large-positive-time regime, $V_{p,\sigma}^{\text{xc}}(t)$ converges to $\omega_p + \mathcal{C} = -i T_K$, meaning that the Kondo resonance peak is created at $\omega = 0$ and it requires no energy transfer. In the small-positive-time regime, $V_{p,\sigma}^{\text{xc}}(t)$ is dominated by the complex constant $\mathcal{C}$, giving rise to the Hubbard side-band. Upon closely comparing our results with NRG, we notice that in the Kondo regime (small $\Gamma/U$), our Kondo peaks have a smaller width than those from NRG. This is because we assume $\Gamma_K\big|_{T=0} = T_K$, while in NRG-based theory, $\Gamma_K\big|_{T=0} \approx 3.92 \, T_K$ [49]. At zero-$T$ and in the WBL, most of the Ansatz parameters can naturally be determined based

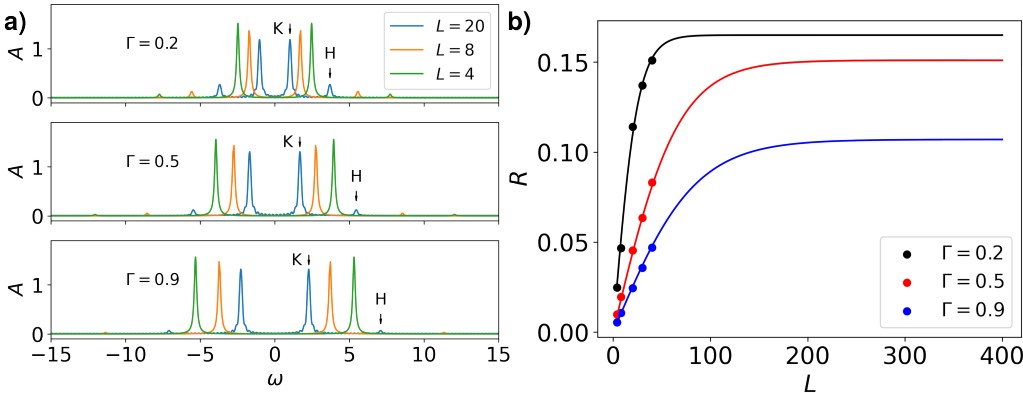

Figure 3: **a)** The zero-$T$ spectral function of $L$-site clusters. We use $U = 3, t_h = 50$ to approach the WBL, and different $V$ values to reach $\Gamma = 0.2, 0.5$ and $0.9$. The results are from ED for $L = 4, 8$ and from TDVP for $L = 20$. The Kondo peaks and Hubbard peaks are highlighted by arrows. **b)** The relative weight $R$ between the Hubbard peak and the Kondo peak, as a function of the total number of sites $L$. TDVP is used for $L = 20, 30$ and $40$. For each value of $\Gamma$, the data are fitted using $R = R_\infty \tanh(aL)$, where $R_\infty$ is the converged value and $a$ is a parameter determining the converging speed. The fitted values are $R_\infty = 0.165, 0.151$, and $0.107$ for $\Gamma = 0.2, 0.5$, and $0.9$, respectively.

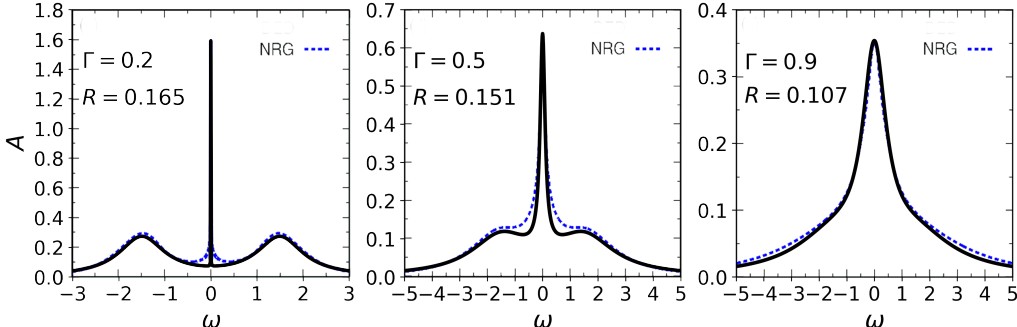

Figure 4: The zero-$T$ spectral function of a symmetric SIAM. We use $U = 3, t_h = 50$ to approach the WBL, and different $V$ values to realize desired $\Gamma$ values. From the extrapolation, we get $\text{Im}[\mathcal{C}] = -0.6, -1.3$ and $-1.7$, respectively, for $\Gamma = 0.2, 0.5$ and $0.9$. The NRG results (blue dashed lines) are adapted from Ref. [29].

on some well-known results of the SIAM. Only one parameter requires a numerical extrapolation. Moreover, the cluster spectral function used in the extrapolation (obtained via ED or TDVP) is actually distinct from the SIAM spectral function: for cluster results, the Kondo peak position is not at $\omega = 0$, and the Hubbard band is too sharp. As already noted close to the end of section 3.2, the discrepancy can be attributed to the essential differences between a finite cluster with tens of sites and a continuous bath. However, the Vxc scheme produces favourable spectral functions using these finite cluster results. This indicates that the Vxc formalism, originating from very fundamental physics and using established knowledge of the target system as a reference, is able to capture the key features of the impurity problem.

Lastly, we discuss the spectral function at finite temperatures. In the Vxc formalism, we can see from the dimer result that thermal excitation leads to the broadening of both the Kondo peak and the Hubbard side-band peak. For the SIAM in the WBL, as the temperature $T$ increases, the Kondo peak is broadened, while the Hubbard side-band remains almost un-

changed. This thermal behavior can be effectively captured by our Ansatz, which treats the imaginary part of the excitation energy $\omega_p$ as temperature-dependent, while assuming that other parameters remain temperature-independent. The temperature dependence can be expressed as

$$\omega_p(T) = \omega_p(T = 0) + i\Omega_T \,. \tag{47}$$

According to our interpretation of the Ansatz parameters, $\omega_p$ at finite-$T$ corresponds to the finite-$T$ Kondo peak half-width $\Gamma_K$ as

$$\omega_p(T) = -\frac{U}{2} + i\big[\Gamma_H - \Gamma_K(T)\big]\,. \tag{48}$$

Following Eqs. (47) and (48), we have the following relation between $\Omega_T$ and $\Gamma_K(T)$:

$$\Omega_T = -i\big[\Gamma_K(T) - \Gamma_K(0)\big]\,. \tag{49}$$

Several expressions have been proposed to describe the temperature-dependence of the Kondo peak half-width $\Gamma_K$ in the literature [50–52], which can be used to determine $\Omega_T$. For $T \lesssim T_K$, an expression beyond Fermi-liquid theory has been derived [52]:

$$\Gamma_K(T) = 1.542 T_K \sqrt{(1 + \sqrt{3}) + (2 + \sqrt{3})\sqrt{1 + \left(\frac{T}{\tilde{T}_K}\right)^2} + \frac{\sqrt{3}}{2}\left(\frac{T}{\tilde{T}_K}\right)^2}\,, \tag{50}$$

where $\tilde{T}_K \approx 0.491 T_K$. Note that Eq. (50) is based on the NRG correction $\Gamma_K(T = 0) \approx 3.92 T_K$, which provides a good description for the SIAM in the strong Kondo regime. However, when the hybridization $\Gamma$ becomes large, the factor 3.92 overestimates the width of the Kondo peak (see the $\Gamma = 0.9$ case in Fig. 4). To be consistent with our zero-$T$ treatment, we rescale Eq. (50) by a factor of 3.92. In the Fermi-liquid regime ($T \ll T_K$), we approximately have

$$\Omega_T = -\frac{\alpha T^2}{T_K}\,, \tag{51}$$

where $\alpha \approx 3.44$. Other Ansatz parameters are estimated using the zero-$T$ TDVP approach. The finite-$T$ spectral function results ($\frac{U}{2t_h} = 2 \times 10^{-3}, \Gamma = 0.04U$) are shown in Fig 5. Compared with NRG results [48], the finite-$T$ Vxc result captures the correct trend of the Kondo peak width: at $T \ll T_K$, the contribution of $\Omega_T$ is negligible, leading to a width dominated by the Kondo temperature. As $T$ approaches $T_K$, $|\Omega_T|$ increases. The agreement with NRG results worsens for $T > 10 T_K$. This may be due to our reference expression, Eq. (50), being only valid for temperatures up to around $T_K$ [52].

## 4 Conclusions and outlook

In this work, we applied the exchange-correlation (xc) field formalism to the symmetric single impurity Anderson model (SIAM) at low temperatures. The formalism introduces a dynamical xc field (Vxc), which can be interpreted as the Coulomb potential of the xc hole. For the SIAM, the Vxc also incorporates the hybridization effect between the impurity and the bath. We proposed an Ansatz for the SIAM Vxc, which exhibits different asymptotic behaviors in the small- and large-time regimes, respectively. At small $t$, the Vxc includes a dominant complex constant term, $\mathcal{C}$, and an exponential term with a complex quasiparticle-like excitation, $\omega_p$. The real and imaginary parts of $\mathcal{C}$ correspond to the peak location and the width of the Hubbard side-band, respectively. At large $t$, the Vxc converges to $\mathcal{C} + \omega_p$, which corresponds to the Kondo

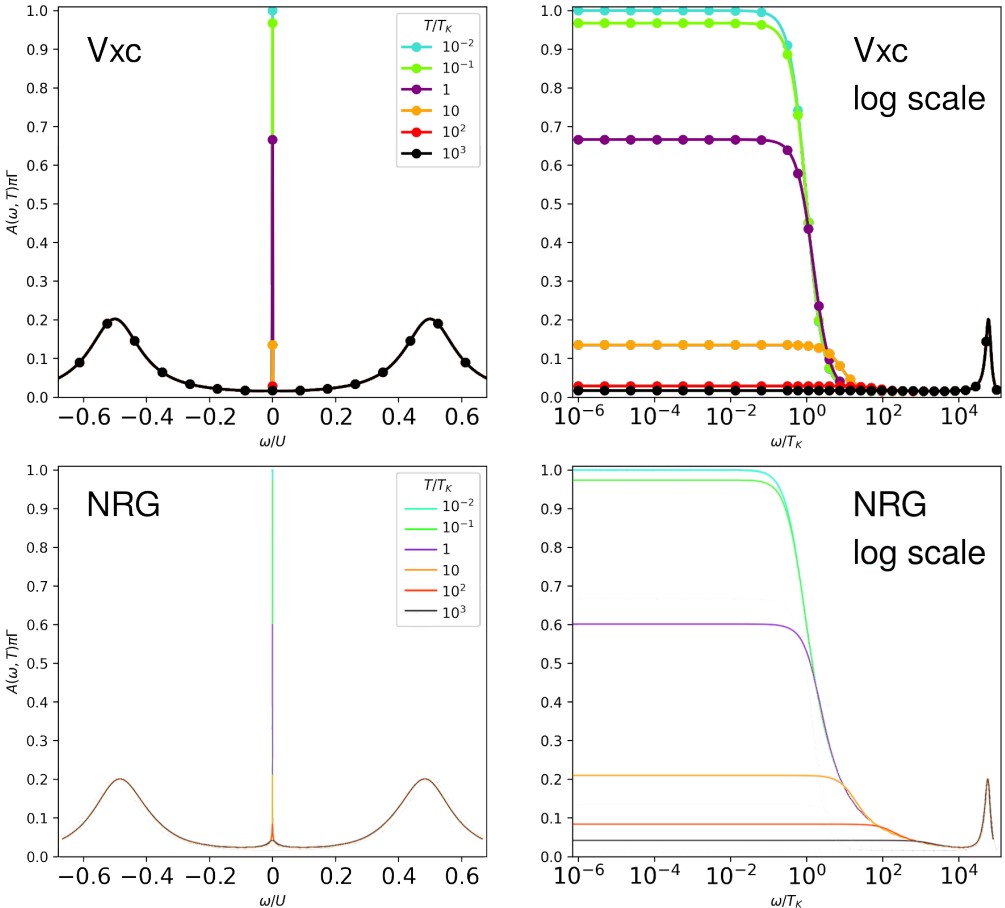

Figure 5: The spectral function of a symmetric SIAM with $\frac{U}{2t_h} = 2 \times 10^{-3}$ and $\Gamma = 0.04U$, at temperatures from $10^{-2}T_K$ to $10^3 T_K$. Left: The frequency is in unit of $U$. Right: The frequency is in unit of $T_K$ and in logarithmic scale to highlight the width of the Kondo resonance peak. The NRG results (bottom) are adapted from Ref. [48].

peak half-width. Importantly, $\text{Im}[\omega_p(T = 0)]$ accounts for the Kondo temperature. At zero-$T$ in the WBL, most parameters of the Ansatz can be calculated from the model parameters using Fermi-liquid theory. The only unknown parameter can be estimated by an extrapolation procedure. For low temperatures, the temperature-dependence of the Ansatz parameters is primarily through $\text{Im}[\omega_p]$, which is determined by extensions of Fermi-liquid theory, guided by the insights from the auxiliary analytically dimer Vxc. Overall, the spectral function calculated from the Vxc shows satisfactory agreement with the NRG results. The extrapolation procedures involved are of low computational cost. We understand the favourable performance of the xc field formalism as follows: the screening effect underlying the SIAM is essential for the Kondo effect, and the xc field provides a suitable description for quasiparticle-like excitations. Hence, the parameters in the Ansatz have clear physical meaning and can be related in a novel perspective to key well-understood features of the spectral function. The fact that only a few parameters require numerical treatment leads to a good trade-off between accuracy and computational effort.

As an outlook, QIMs beyond the symmetric SIAM at half-filling can also be interpreted from the perspective of the xc field formalism. We have already noted that for the narrow band SIAM or the SIAM away from half-filling, the Vxc requires a different extrapolation scheme. Additionally, when an external magnetic field is included, the spectral function becomes spin-dependent, and the Kondo resonance can be suppressed with increasing field. In future work, we plan to investigate how the magnetic field affects the Vxc. Moreover, quantities such as the dynamical spin susceptibility, the specific heat, and the size of the Kondo cloud are related to spin correlators. We expect the spin xc field formalism can be applied to these problems.

Finally, we stress a significant feature of the xc field formalism: it manages to reduce a complicated many-body problem to an extrapolation procedure. The extrapolation is usually done with a (numerically) solvable finite cluster or a homogeneous system as a reference. When the target system and the reference system exhibit explicit similarities, the extrapolation can be done straightforwardly. In practice, the connection between the reference system and the complex target is often less obvious. An example is the SIAM presented in this paper, where the finite cluster spectral function differs qualitatively from the SIAM. Despite this, the xc field formalism successfully captures the implicit correspondence, specifically the relative weight between the Hubbard peak and the Kondo peak at $T = 0$. Hence, we believe that the xc field formalism, based on the quasiparticle picture, is a viable and powerful approach for modeling correlated many-body system and holds great potential for first-principles calculations.

## Acknowledgments

I would like to thank Ferdi Aryasetiawan and Claudio Verdozzi for stimulating discussions, reading the manuscript and providing useful comments.

**Funding information**   I acknowledge support from the Swedish Research Council (grants number 2017-03945 and 2022-04486).

## A   Low temperature approximation of the Vxc

From Eq. (27) and $e^{-\beta E_m} \ll 1$ for $m > 2$, the Vxc can be written as

$$V_{p,\sigma}^{\text{xc}}(t,\beta) = \frac{C + e^{-\beta \Delta_0} D}{A + e^{-\beta \Delta_0} B} \,, \tag{A.1}$$

where

$$\Delta_0 = E_2 - E_1 \,, \tag{A.2}$$

$$A = \sum_{n_+} a_{n_+,1} e^{-i\omega_{n_+,1} t} \,, \tag{A.3}$$

$$B = \sum_{n_+} a_{n_+,2} e^{-i\omega_{n_+,2} t} \,, \tag{A.4}$$

$$C = \sum_{n_+} a_{n_+,1} \omega_{n_+,1} e^{-i\omega_{n_+,1} t} \,, \tag{A.5}$$

$$D = \sum_{n_+} a_{n_+,2} \omega_{n_+,2} e^{-i\omega_{n_+,2} t} \,. \tag{A.6}$$

An expansion

$$\frac{1}{A + e^{-\beta \Delta_0} B} \approx \frac{1 - e^{-\beta \Delta_0 \frac{B}{A}}}{A}, \tag{A.7}$$

leads to

$$V_{p,\sigma}^{\text{xc}}(t,\beta) = \frac{C}{A} + \Big[\frac{D}{A} - \frac{BC}{A^2}\Big]e^{-\beta \Delta_0}, \tag{A.8}$$

where $C/A$ is just the zero-$T$ Vxc, and all terms of order $\mathcal{O}(e^{-\beta \Delta})$ where $\Delta > \Delta_0$ are neglected for low-$T$. The time-oscillating term in the main text is then

$$\tilde{V}_{p,\sigma}(t) = \frac{D}{A} - \frac{BC}{A^2} = \frac{C}{A}\Big[\frac{D}{C} - \frac{B}{A}\Big]. \tag{A.9}$$

## B  Analytic Vxc of an impurity-free-electron dimer

We use a dimer consisting of an interacting site ($f$) and a noninteracting site ($c$) to calculate the analytic Vxc for the $f$ site. Here we repeat the Hamiltonian in the main text:

$$\hat{H}_{\text{dimer}} = \epsilon_f (\hat{n}_{f\uparrow} + \hat{n}_{f\downarrow}) + U\hat{n}_{f\uparrow}\hat{n}_{f\downarrow} + V\sum_{\sigma}(\hat{f}_{\sigma}^{\dagger}\hat{c}_{\sigma} + \text{H.c.}), \tag{B.1}$$

where $\epsilon_f = -\frac{U}{2}$, $V$ is the hopping strength and the dimer is at half-filling. With the chosen dimer model, the spectral weights and the local Green's function (Eq. (21)) are spin-independent. For simplicity we keep the spin indices implicit in this section. With variables depending on $U, V$

$$u = \frac{U}{2V}, \tag{B.2}$$

$$x = \frac{u}{4} + \sqrt{1 + \Big(\frac{u}{4}\Big)^2}, \tag{B.3}$$

$$y = \frac{u}{2} + \sqrt{1 + \Big(\frac{u}{2}\Big)^2}, \tag{B.4}$$

and the same low-$T$ assumption, the particle part of the Green's function can be written as

$$i\bar{G}_{ff}^{p}(t) = Z^{-1}\Big[\big[a_{1,1}e^{-i\omega_{1,1}t} + a_{2,1}e^{-i\omega_{2,1}t}\big] + e^{\beta u}\big[a_{1,2}e^{-i\omega_{1,2}t} + a_{2,2}e^{-i\omega_{2,2}t}\big]\Big], \tag{B.5}$$

where the partition function is

$$Z = 1 + 3e^{\beta(u-2x)V}, \tag{B.6}$$

the spectral weights are

$$a_{1,1} = \frac{(x+y)^2}{2(1+x^2)(1+y^2)}, \tag{B.7}$$

$$a_{2,1} = \frac{(1-xy)^2}{2(1+x^2)(1+y^2)}, \tag{B.8}$$

$$a_{1,2} = \frac{3}{2(1+y^2)}, \tag{B.9}$$

$$a_{2,2} = \frac{3y^2}{2(1+y^2)}, \tag{B.10}$$

and the excitation energies are

$$\omega_{1,1} = (2x - y)V, \tag{B.11}$$

$$\omega_{2,1} = (2x + y - u)V, \tag{B.12}$$

$$\omega_{1,2} = (-y + u)V, \tag{B.13}$$

$$\omega_{2,2} = yV. \tag{B.14}$$

Following the notations in Appendix A, the zero-$T$ Vxc is

$$V_p^{\text{xc}}(t, T = 0) = \frac{a_{1,1}\omega_{1,1}e^{-i\omega_{1,1}t} + a_{2,1}\omega_{2,1}e^{-i\omega_{2,1}t}}{a_{1,1}e^{-i\omega_{1,1}t} + a_{2,1}e^{-i\omega_{2,1}t}}. \tag{B.15}$$

We note that in the large-interaction regime ($u \gg 1$),

$$x = \frac{u}{2} + \frac{2}{u} + \mathcal{O}\left(\frac{1}{u^3}\right), \tag{B.16}$$

$$y = u + \frac{1}{u} + \mathcal{O}\left(\frac{1}{u^3}\right), \tag{B.17}$$

thus

$$\lambda := \frac{a_{1,1}}{a_{2,1}} = \frac{9}{u^2} + \mathcal{O}\left(\frac{1}{u^3}\right) \ll 1. \tag{B.18}$$

The expansion of Vxc to the first order in $\lambda$ gives

$$V_p^{\text{xc}}(t, T = 0) \approx \omega_{2,1} - \lambda \Omega e^{i\Omega t}, \tag{B.19}$$

where

$$\Omega = \omega_{2,1} - \omega_{1,1} = \sqrt{u^2 + 4}V. \tag{B.20}$$

For low-$T$, following Eq. (A.9), we have

$$\frac{\tilde{V}_p(t)}{V_p^{\text{xc}}(t, T = 0)} = \frac{a_{1,2}\omega_{1,2}e^{-i\omega_{1,2}t} + a_{2,2}\omega_{2,2}e^{-i\omega_{2,2}t}}{a_{1,1}\omega_{1,1}e^{-i\omega_{1,1}t} + a_{2,1}\omega_{2,1}e^{-i\omega_{2,1}t}} - \frac{a_{1,2}e^{-i\omega_{1,2}t} + a_{2,2}e^{-i\omega_{2,2}t}}{a_{1,1}e^{-i\omega_{1,1}t} + a_{2,1}e^{-i\omega_{2,1}t}}. \tag{B.21}$$

Noting that for $u \gg 1$,

$$\frac{a_{1,2}}{a_{2,1}} = \frac{3}{u^2} + \mathcal{O}\left(\frac{1}{u^3}\right), \tag{B.22}$$

$$\frac{a_{2,2}}{a_{2,1}} = 3\left(1 + \frac{4}{u^2}\right) + \mathcal{O}\left(\frac{1}{u^3}\right), \tag{B.23}$$

$$\frac{\omega_{1,1}}{\omega_{2,1}} = \frac{3}{u^2} + \mathcal{O}\left(\frac{1}{u^3}\right), \tag{B.24}$$

$$\frac{\omega_{1,2}}{\omega_{2,1}} = -\frac{1}{u^2} + \mathcal{O}\left(\frac{1}{u^3}\right), \tag{B.25}$$

$$\frac{\omega_{2,2}}{\omega_{2,1}} = 1 - \frac{4}{u^2} + \mathcal{O}\left(\frac{1}{u^3}\right), \tag{B.26}$$

we get

$$\frac{\tilde{V}_p(t)}{V_p^{\text{xc}}(t, T = 0)} = \frac{24}{u^2}e^{i\Omega' t} - \frac{12}{u^2}e^{i\Omega'' t} + \mathcal{O}\left(\frac{1}{u^3}\right), \tag{B.27}$$

where

$$\Omega' = \omega_{2,1} - \omega_{1,2} = \left( \sqrt{\frac{u^2}{4} + 4} + \sqrt{u^2 + 4} - \frac{u}{2} \right) V, \tag{B.28}$$

$$\Omega'' = \omega_{2,1} - \omega_{2,2} = \left( \sqrt{\frac{u^2}{4} + 4} - \frac{u}{2} \right) V. \tag{B.29}$$

The temperature factor,

$$e^{-\beta \Delta_0} = e^{-\beta(2x-u)}, \tag{B.30}$$

need to be small in order for the approximation Eq. (A.7) holds.

## C Calculating the Green's function at $T = 0$ using TDVP

Here, we give some details regarding calculating $G_{ff,\sigma}(t, T = 0)$ of our finite cluster using the ITensor library [43, 44]. We first calculate the ground-state $|\Psi_0\rangle$ using the density matrix renormalization group algorithm. Then the TDVP algorithm is used to time-evolve the state $\hat{f}_\sigma^\dagger |\Psi_0\rangle$. With $|\Psi(t > 0)\rangle = e^{-i\hat{H}t} \hat{f}_\sigma^\dagger |\Psi_0\rangle$, the one-particle Green's function at equilibrium can be calculated:

$$\begin{aligned} iG_{ff,\sigma}(t > 0) &= \langle \Psi_0 | e^{i\hat{H}t} \hat{f}_\sigma e^{-i\hat{H}t} \hat{f}_\sigma^\dagger | \Psi_0 \rangle \\ &= e^{iE_0 t} \langle \Psi_0 | \Psi(t) \rangle, \end{aligned} \tag{C.1}$$

where $E_0$ is the ground-state energy. Our system is particle-hole symmetric, which means

$$G_{ff,\sigma}(t < 0, T = 0) = -G_{ff,\sigma}(-t, T = 0). \tag{C.2}$$

We calculate the Green's function in the frequency domain with the Fourier transform

$$G_{ff,\sigma}(\omega, T = 0) = \int G_{ff,\sigma}(t, T = 0) e^{i\omega t} dt. \tag{C.3}$$

The spectral function is then calculated from $G_{ff,\sigma}(\omega)$.

## D Solving the Green's function from the Ansatz of the Vxc

For the SIAM, the equation of motion of the particle Green's function ($t > 0$) reads

$$[i\partial_t - \epsilon_f - V^H - V_{p,\sigma}^{xc}(t, \beta)] \bar{G}_{ff,\sigma}^p(t, \beta) = 0, \tag{D.1}$$

where $\epsilon_f + V^H = 0$ and $g^+ = \bar{G}_{ff,\sigma}^p(t = 0^+, \beta) = -0.5i$ for the symmetric SIAM. Accordingly, the Green's function is

$$\bar{G}_{ff,\sigma}^p(t > 0, \beta) = g^+ e^{-i \int_0^t V_{p,\sigma}^{xc}(\bar{t}, \beta) d\bar{t}}. \tag{D.2}$$

The Vxc is given by the complete Ansatz

$$V_{p,\sigma}^{xc}(t > 0, \beta) = \frac{\lambda(\mathcal{C} + \omega_p) + (1 - \lambda)\mathcal{C} e^{i\omega_p t}}{\lambda + (1 - \lambda) e^{i\omega_p t}}, \tag{D.3}$$

where $\lambda$ is real and positive, and $\omega_p$ and $\mathcal{C}$ are complex. Assuming that $\lambda \ll 1$ and $\mathrm{Im}\,\omega_p > 0$, we have for positive $t$

$$V_{p,\sigma}^{\mathrm{xc}}(t,\beta) = \begin{cases} \lambda\omega_p e^{-i\omega_p t} + \mathcal{C}, & t \text{ small,} \\ \omega_p + \mathcal{C}, & t \text{ large.} \end{cases} \tag{D.4}$$

The time integral of $V^{\mathrm{xc}}$ is

$$\begin{aligned} \int_0^t V_{p,\sigma}^{\mathrm{xc}}(\bar{t},\beta)d\bar{t} &= \int_0^t \frac{\lambda(\mathcal{C}+\omega_p)e^{-i\omega_p\bar{t}}}{\lambda e^{-i\omega_p\bar{t}} + (1-\lambda)}d\bar{t} + \int_0^t \frac{(1-\lambda)\mathcal{C}e^{i\omega_p\bar{t}}}{\lambda + (1-\lambda)e^{i\omega_p\bar{t}}}d\bar{t} \\ &= \frac{\mathcal{C}+\omega_p}{-i\omega_p}\ln\left[\lambda e^{-i\omega_p t} + (1-\lambda)\right] + \frac{\mathcal{C}}{i\omega_p}\ln\left[\lambda + (1-\lambda)e^{i\omega_p t}\right] \\ &= (\mathcal{C}+\omega_p) + i\ln\left[\lambda + (1-\lambda)e^{i\omega_p t}\right]. \end{aligned} \tag{D.5}$$

Applying Eq. (D.5) to Eq. (D.2), we obtain the Green's function

$$\bar{G}_{ff,\sigma}^p(t>0,\beta) = g^+\left[(1-\lambda)e^{-i\mathcal{C}t} + \lambda e^{-i(\mathcal{C}+\omega_p)t}\right]. \tag{D.6}$$

The parameters are interpreted as follows: $\mathrm{Im}[\mathcal{C}] \sim -\Gamma_{\mathrm{H}}$, where $\Gamma_{\mathrm{H}}$ is the half-width of the Hubbard side-band, and $\mathrm{Im}[\mathcal{C}+\omega_p] \sim -\Gamma_{\mathrm{K}}$, where $\Gamma_{\mathrm{K}}$ is the half-width of the Kondo peak and is much smaller than $\Gamma_{\mathrm{H}}$ in the Kondo regime. Using the results in the main text, we have $\lambda \ll 1, \mathcal{C}+\omega_p = -iT_{\mathrm{K}}$, and $\omega_p = -\frac{U}{2} + i(\Gamma_{\mathrm{H}} - \Gamma_{\mathrm{K}})$. They are consistent with our assumption to derive the asymptotic properties in Eq. (D.4).

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
