# Peer review of "Kondo spectral functions at low-temperatures: A dynamical-exchange-correlation-field perspective."

_SciPost Physics, doi:SciPost Phys. 18, 002 (2025)_

## Round 1 · Referee Report · Anonymous (Referee 1) · 2024-9-16

Report

The present manuscript applies the formalism of the so-called
"dynamical exchange-correlation (xc) field" of Ref.[37] to the single-impurity Anderson model. The authors proposes a rather simple ansatz (Eq.(30)) for this dynamical xc field to obtain the spectral function of the Anderson model in the Kondo regime where the parameters are fixed by using the known peak positions and widths of the spectral peaks. This ansatz seems to work surprisingly well given its simplicity.

While I find the paper interesting in general, I still have a number of
points which I would like the author to address:

1. The coefficient a_{n+,m} is defined just after Eq.(14). Shouldn't this also be sigma-dependent?

2. On Eq.(20): first, I suppose it is only meant to be valid for t>0, no?
Second, I am a bit confused about its form: why is there no explicit
dependence on the interaction U? Shouldn't it (loosely speaking) be
something like U G^(2)(t)/G(t) where G^(2) is the two-particle Green
function? Also, I don't understand the factor \aN{n+,m} \omega_{n^+,m} in the denominator. I would have expected this to be
<m| \hat{n}_{-\sigma} f_{\sigma} |n+><n+|f^{\dagger}_{\sigma} |m>.
Please clarify!

3.Please give more details on what is actually done in Sec. 3.2 and how, such that interested readers could repeat the calculations. The time-dependent variational principle is used to obtain which quantity, the one-particle Green function of the cluster?

4.In Fig.4: could the author plot the NRG results on top of the present results for better comparison? The same applies for Fig.5.

5.In Eq.(37): I assume that the parameter \Omega_T is temperature dependent? How is this parameter determined in practice? Is it used as a fit parameter to reproduce known spectral functions? Please show its evolution as function of temperature!

6.Finally, I noticed a typo in line 141: it should be "emphasize" instead of
"emphasis"

To summarize, before I can recommend this manuscript for publication in SciPost Physics I would like to see the issues raised above being addressed.

Recommendation

Ask for major revision

  • validity: -
  • significance: -
  • originality: -
  • clarity: -
  • formatting: -
  • grammar: -

Author:  Zhen Zhao  on 2024-10-25  [id 4901]

(in reply to Report 1 on 2024-09-16)

I would like to thank the referee for the report. Please find the "ResponseSciPost.pdf" document that answers directly to both referees' questions and suggestions. A revised manuscript will be submitted with a formal list of changes.

Attachment:

ResponseSciPost_hDrZXOF.pdf

---

## Round 1 · Referee Report · Anonymous (Referee 2) · 2024-9-18

Report

This paper applies a novel method, the so called "dynamical exchange-correlation field" (Ref. 37), to the Anderson impurity model in order to compute spectral functions in the Kondo regime.

I find the paper quite interesting. But I think the paper lacks clarity in a couple of places. The paper would also benefit from better explanations in some parts, since the dynamical exchange-correlation field is a novel approach. I have a couple of comments and questions that the author should address before I can agree to publication:

(1) I was at first confused by Eq. (11): How could the dynamic exchange correlation hole for $r^{\prime\prime}=r$ become time-independent and just equal to the negative density? But this follows from Eq.(8) and the fact that the second order Green's function $G^{(2)}(r,r^\prime,r^{\prime\prime};t)$ (using the notation of Ref. 37) vanishes for $r^{\prime\prime}=r$, and thus also the correlation function $g(r,r^\prime,r;t)=0$. I think the author should give this explanation after Eq. (11) to help the reader.

(2) Eqs. (13) and (14) are the Lehmann representations of the Green's function (which should be mentioned), and the denominators are just the partition functions $Z$. I think the equations would become clearer if $Z$ was introduced and used. In the following equation (20), the denominators cancel anyway.

(3) To help the reader, it should be explicitly stated that Eq. (20) follows from applying the equation of motion (15) to the Lehmann representation and solving for Vxc.

(4) Sec. 3.1, after Eq. (22): I am not sure whether it is appropriate to speak of "Kondo regime" in the context of the Anderson dimer. The Kondo effect is usually associated with an impurity coupled to a continuous band of conduction electrons.

(5) The last two sentences of Sec. 3.1, p. 8: I think this explanation for the temperature induced broadening follows simply from the Lehmann representation of the GF (13,14) which the author used to obtain the approximation for the dynamic Vxc.

(6) Is the Vxc given by Eq. (30) valid only for $t>0$? If so, what is the corresponding equation for $t<0$? I think it would also be interesting to see Vxc in the frequency domain, i.e. the Fourier transform of Eq. (30), which could then be compared to the self-energy for the SIAM. I suspect they must be very similar in the case of the SIAM.

(7) How did the author arrive at the hyperbolic-tangent form for $R(L)$ fitted to the data in Fig. 3b? Is that based on some theoretical background? Otherwise I think the actual functional form cannot be extrapolated from the calculated data, since the data is still largely in the linear regime. Very different functional forms leading to very different limits $R(L=\infty)$could be compatible with the data.

(8) It would be nice if in Figs. 4 and 5 the calculated spectra would be directly compared to the NRG spectra of Refs. [28] and [47].

Recommendation

Ask for major revision

  • validity: -
  • significance: -
  • originality: -
  • clarity: -
  • formatting: -
  • grammar: -

Author:  Zhen Zhao  on 2024-10-25  [id 4900]

(in reply to Report 2 on 2024-09-18)

I would like to thank the referee for the report. Please find the "ResponseSciPost.pdf" document below that answers directly to both referees' questions and suggestions. A revised manuscript will be submitted with a formal list of changes.

Attachment:

ResponseSciPost.pdf

---

## Round 2 · Referee Report · Anonymous (Referee 1) · 2024-11-11

Report

I am happy with the author's reply to my previous report and also
to the report of the other referee and now recommend publication of
this manuscript in SciPost Physics.

Just one tiny detail which needs to be corrected: in line 355, for
the quantity \tilde{T}_K the energy unit is missing which should be T_K, i.e., \tilde{T}_K \approx 0.491 T_K.

Requested changes

1) in line 355, for the quantity \tilde{T}_K the energy unit is missing which should be T_K, i.e., \tilde{T}_K \approx 0.491 T_K.

Recommendation

Publish (easily meets expectations and criteria for this Journal; among top 50%)

  • validity: high
  • significance: high
  • originality: good
  • clarity: good
  • formatting: excellent
  • grammar: good

Author:  Zhen Zhao  on 2024-11-20  [id 4967]

(in reply to Report 1 on 2024-11-11)

I thank the referee for giving positive evaluation for the manuscript and for pointing out the typo.
Eq. (50) is from Eq. (10) of Ref. 52, and as the referee noticed, in line 355, it should be \tilde{T}_K\approx 0.491 T_K.
This typo will be corrected in the revised manuscript.

---

## Round 2 · Referee Report · Anonymous (Referee 2) · 2024-11-19

Report

The author has answered to the comments and questions of both referees satisfactory, and revised the manuscript accordingly. All my doubts have been answered, and I think the manuscript is much clearer now. I am glad that our comments helped the author to improve his Ansatz for the dynamical xc potential of the Anderson model. The results are very nice and compare very well to NRG data.

However, I have noticed two mistakes introduced in the newly added part in Sec. 3.3 related to the width of the Kondo peak and its relation to the Kondo temperature (see below for details). These mistakes are not essential, but should be corrected before the paper can be published, in order to avoid confusion. I therefore recommend publication after these mistakes have been corrected.

Requested changes

1) The first mistake concerns the relation between the Kondo peak width at T=0, and the Kondo temperature (lines 329-330). The proportionality factor ~3.7 was found numerically from NRG calculations in Ref. 49. But an exact analytic relationship in terms of the Wilson number was given in Ref. 52 which results in a proportionality factor of 3.92, close to the numerical one. In this context it is also confusing to use w=3.7, since "w" usually would denote the Wilson number (w=0.4128).

2) This is probably related to the first one: Eq. (50) reproduces Eq. (10) of Ref. 52, but with $\tilde{T}_K$ instead of $\Delta_K$ (the Frota width parameter). It is not clear to me what is $\tilde{T}_K$ and why can it be set to ~0.491?! Is it a typo? The value of $\Delta_K$ is linked to the Kondo temperature by $\Delta_K=1.542\,T_K$ (as correctly used in the prefactor to the square-root in Eq. (50)), and therefore depends on the interaction U and the bare width $\Gamma$. Also as stated above $\Gamma_K(T=0)\approx3.92\,T_K$, so the rescaling should be done with 3.92 instead of 3.7, as stated after Eq. (50). Maybe this rescaling is also somewhat confusing, and the author should think about using $T_K$ as the Kondo temperature (as defined by Wislon), and use $\Gamma_K^0$ for the Kondo peak width.

Recommendation

Ask for minor revision

  • validity: high
  • significance: high
  • originality: high
  • clarity: high
  • formatting: excellent
  • grammar: excellent

Author:  Zhen Zhao  on 2024-11-20  [id 4968]

(in reply to Report 2 on 2024-11-19)

I thank the referee for the positive evaluation for the manuscript and for pointing out the typo and the statement that may lead to confusion.

  1. Regarding the first point, we followed Eq. (10) of Ref. 52 to determine the relation between the Kondo temperature $T_K$ and the Kondo peak half-width $\Gamma_K$, which means only the factor 3.92 is considered. We agree that mentioning the numerical factor 3.7 and using the symbol "w" may cause confusion. These issues will be corrected in the revised manuscript.

  2. Concerning the second point, following Eq. (10) of Ref. 52, it should be $\tilde{T}_K\approx1.542T_K/\pi=0.491T_K$. We apologize for the typo. As mentioned in the first point, we have used the factor 3.92 for the rescaling. By comparing with NRG results, we observe that $\Gamma_K^0=3.92 T_K$ is a good description for the SIAM in the strong Kondo regime, but not when $\Gamma$ is large. Therefore, we used $\Gamma_K^0\sim T_K$ instead of $\Gamma_K^0=3.92 T_K$ for our zero-$T$ treatment. This clarification will be included in the revised manuscript to explain our rescaling approach.

Finally, I wish to thank the referee again for his/her remarks and suggestions. I hope to have properly addressed them and clarified the manuscript.

---

## Round 2 · Author Response

Dear Editors,

I herewith resubmit the manuscript "Kondo spectral functions at low-temperatures: A dynamical-exchange-correlation-field perspective" for publication in SciPost Physics.

I would like to thank the referees for their insightful reports. The referees provided general observations as well as detailed questions, some of which helped us improve our original approach and inspired us to deepen our understanding of the results. In accordance with our replies sent to the referee reports, text changes have been made in the revised manuscript, addressing the referees' observations and suggestions to make the manuscript more suitable for publication in SciPost Physics.

Finally, I would like to thank you for your consideration and handling of this manuscript.

With best regards,
Z. Zhao

---

## Round 2 · List of Changes

1. Changes in Sec. 2
A relation between the dynamical xc field and the self-energy in frequency domain has been added.

An explanation of the exact constraint of the dynamical xc hole has been included.

2. Changes in Sec. 2.1
The use of the Lehmann representation of the Green's function is now explicitly pointed out.

Notations related to spin indices have been revised for consistency through the manuscript.

3. Changes in Sec. 2.2
It is now explicitly stated that the xc field (Eq. 27 in the revised manuscript) is derived by applying the Lehmann representation of the Green's function to its equation of motion and solving for the xc field.

4. Changes in Sec. 3.1
For the dimer, the term "Kondo regime" has been replaced with "large interaction regime" for precision.

The statement in the end of this subsection regarding the temperature-induced broadening of the spectral peaks has been revised.

5. Changes in Sec. 3.2
In the end of this subsection, additional discussions have been added regarding the fact that finite cluster results provide insights about the xc field in the small-time regime. For the SIAM, the Kondo resonance is related to the large-time behavior of the xc field.

6. Changes in Sec. 3.3
An ansatz capturing both small- and large-time behavior of the xc field has been proposed. The Green's function can now be calculated by directly integrating the ansatz over time, which improves the treatment in the original manuscript, where a low-order expansion was applied.

The way of calculating the xc field with negative time using the particle-hole symmetry is now explicitly described.

Interpretations of the ansatz parameters (Eqs. 43-45 in the revised manuscript) are explained in more details.

Additional comments on the hyperbolic-tangent functional fitting are provided.

The treatment of the ansatz parameters at low-temperatures (Eqs. 47-51 in the revised manuscript) is now improved. A Fermi-liquid-theory-based expression for the Kondo peak half-width in the literature is now used as reference.

NRG results adapted from some literature have been added in Fig. 4 and Fig.5 for better comparison. With the improved treatment at low-temperatures, we now plot the spectral functions in Fig. 5 for additional temperature values. We also expand discussions about the detailed difference between our results and those from NRG.

7. Changes in Sec. 4
Discussions on the asymptotic behaviors of the xc field in the small- and large-time regimes have been added.

8. Changes in Appendix C
A detailed explanation of how the Green's function on a finite cluster at zero-temperature can be calculated using the TDVP method is provided.

9. Changes in Appendix D
We derive the Green's function using the improved ansatz of the xc field, showing that no low-order expansion is involved.

10. Changes in References
Additional citations have been added regarding the reference expression for the Kondo peak half-width in Sec. 3.3.

---

## Round 3 · Author Response

Dear Editor,

Here I resubmit the manuscript following the referees' suggestions. The changes are minor, which include corrections of typos and short clarification of the approach. I hope the manuscript is now considered suitable for publication in SciPost Physics.

With my best regards,
Z. Zhao

---

## Round 3 · List of Changes

1. Line 355. The typo is corrected: \tilde{T}_k=0.491 -> \tilde{T}_K=0.491 T_K.
  2. Line 355-359. Texts added to avoid confusion and to clarify our rescaling treatment.

---

## Editorial Decision

published